# Physical Exercise Promotes DNase Activity Enhancing the Capacity to Degrade Neutrophil Extracellular Traps

**DOI:** 10.3390/biomedicines10112849

**Published:** 2022-11-08

**Authors:** Anna S. Ondracek, Adrienne Aszlan, Martin Schmid, Max Lenz, Andreas Mangold, Tyler Artner, Michael Emich, Monika Fritzer-Szekeres, Jeanette Strametz-Juranek, Irene M. Lang, Michael Sponder

**Affiliations:** 1Department of Internal Medicine II, Division of Cardiology, Medical University of Vienna, 1090 Vienna, Austria; anna.ondracek@meduniwien.ac.at (A.S.O.); adrienne.aszlan@meduniwien.ac.at (A.A.); martin.g.schmid@hotmail.com (M.S.); max.lenz@meduniwien.ac.at (M.L.); andreas.mangold@lkhf.at (A.M.); tyler.artner@meduniwien.ac.at (T.A.); irene.lang@meduniwien.ac.at (I.M.L.); 2Austrian Federal Ministry of Defence, Austrian Armed Forces, 1090 Vienna, Austria; dr@emich.at; 3Department of Laboratory Medicine, Medical University of Vienna, 1090 Vienna, Austria; monika.fritzer-szekeres@meduniwien.ac.at; 4Rehabilitation Centre Bad Tatzmannsdorf, 7431 Bad Tatzmannsdorf, Austria; dr.jstrametz@sternvilla.com

**Keywords:** physical exercise, DNase activity, neutrophil extracellular traps, cardiovascular health, cardiovascular diseases

## Abstract

(1) Background: An unhealthy lifestyle is a significant contributor to the development of chronic diseases. Physical activity can benefit primary and secondary prevention. Higher DNase activity is associated with favourable outcomes after cardiovascular (CV) events. In this study, we aimed to investigate the influence of consequent endurance exercise on DNase activity. (2) Methods: 98 subjects with at least one CV risk factor but the physical ability to perform endurance training were included. Individuals performed a bicycle stress test at the beginning and after 8 months to assess physical performance. In between, all participants were instructed to engage in guideline-directed physical activity. Blood samples were drawn in two-month intervals to assess routine laboratory parameters, cell-free DNA (cfDNA), and DNase activity. (3) Results: Prevailing CV risk factors were overweight (65.9%), a positive family history (44.9%), hypertension (32.7%) and smoking (20.4%). Performance changed by 7.8 ± 9.1% after 8 months. Comparison of baseline to 8 months revealed a decrease in cfDNA and an increase in DNase activity. This effect was driven by participants who achieved a performance gain. (4) Conclusions: Regular physical activity might improve CV health by increasing DNase activity and thereby, the capacity to lower pro-inflammatory signalling, complementing measures of primary and secondary prevention.

## 1. Introduction

Chronic conditions, such as cardiovascular disease (CVD), cancer, obesity or diabetes, are major causes of mortality [1,2,3]. Contributors to the development of chronic disease are often lifestyle-related factors, such as abuse of tobacco and alcohol, diets containing high amounts of salt and fat, as well as physical inactivity. A healthy lifestyle represents a non-pharmacological intervention, which can substantially prevent or reduce chronic disease burden [4].

In 2010, the World Health Organization stated physical inactivity as the fourth-leading cause of death worldwide [5], only to report no improvement in global levels ten years later [6], despite the well-known health advantages of sports. By inducing a complex multisystem response, an array of cells and organs benefit from exercise [7,8,9], next to improving mental health, and cardiorespiratory fitness [10]. Including physical activity as therapeutic modality requires a more profound knowledge of exercise-induced mechanisms on the cellular and molecular system in order to reflect the individual risk profile of patients.

Depending on intensity and duration, acute physical activity induces hemostatic perturbations, including thermal, metabolic, hormonal and mechanical stress, which result in a shift in coagulation system dynamics, mobilization and activation of leukocytes, as well as an increase in generation of reactive oxygen species and acute-phase proteins [11,12]. Catecholamine release causes an increase in vascular permeability and, subsequently, neutrophil counts shortly after physical activity [13]. These metabolic changes are accompanied by a phenomenon usually attributed to a range of inflammatory pathological conditions: an immediate and transient rise in plasma levels of circulating cell-free deoxyribonucleic acid (cfDNA) [14,15,16]. While biological properties, purpose, and origin remain only partially understood, a reasonable explanation emerged with the concept of neutrophil extracellular traps (NETs). Triggered by various stimuli, activated neutrophils can emit their nuclear content into extracellular space (NETosis), creating a web-like meshwork of decondensed double-stranded DNA, decorated with proteins and enzymes originating from granules. Adsorbed antimicrobial molecules degrade pathogens previously caught and immobilized in the expelled chromatin fibers, classifying NETosis as a crucial effector mechanism of immune defense [17]. In addition, NETs have pro-inflammatory [18], cytotoxic [19,20] and pro-thrombotic [21] properties and these characteristics contribute to the development or progression of a variety of diseases, including atherosclerosis [22], autoimmune disease [23], abdominal aortic aneurysms [24], coronavirus disease 2019 [25,26] and sepsis [18]. The balance between NET formation and degradation might regulate their pathological potential. Endogenous counter regulators destructing the reticular structures are desoxyribonucleases (DNases), as they hydrolyse the DNA backbone, leading to removal from the circulation [17]. CfDNA and especially NETs can be immunogenic and elicit pro-inflammatory signaling [27]; hence, intact DNase activity is crucial to prevent chronic inflammation and subsequent auto-immune reactions [28]. Deficient DNase activity and a corresponding relative accumulation of cfDNA have been associated with long-term mortality in a cohort of myocardial infarction patients [29]. In the acute setting, higher DNase activity was associated with lower infarct size, potentially driven by accelerated thrombus lysis, as observed in vitro [30]. The benefits of DNase activity have also been recognized for their therapeutic potential [31], as several clinical trials are currently registered.

Although some evidence already points to the induction of NETs by physical activity [32], most studies investigate short-term consequences of exhaustive exercise [33,34]. To enhance our understanding of the benefits incited by regular endurance exercise, this study aimed to investigate the influence of consequent physical activity over an 8-month time course on levels of cell-free DNA (cfDNA) and DNase activity in a real-world healthy study group.

## 2. Materials and Methods

### 2.1. Study Protocol

Participants were recruited in collaboration with the Austrian Federal Ministry of Defence. In total, 109 staff members volunteered to participate in the study. The premature termination of eleven participants due to personal reasons involving accidents or loss of motivation resulted in the final inclusion of 98 individuals aged between 30 and 65 years with the physical ability to perform endurance exercise. Eligible subjects were additionally required to fulfil the criteria for at least one cardiovascular risk factor, defined as follows:Diagnosed chronic heart disease with prior myocardial infarction, CABG, PCI or stroke.A positive family anamnesis of first-degree relatives (mother or father) regarding CVD or stroke.The presence of one or more metabolic risk factors, including overweight (BMI > 25), diabetes mellitus (HbA1c > 6.5% or the presence of antidiabetic medication), dyslipidemia (marked by statin intake) and arterial hypertension (resting SBP > 140 mmHg/resting DBP > 85 mmHg or the presence of antihypertensive medication).A positive smoking status.

Individuals who were not considered fit for endurance exercise or with increased inflammation parameters indicative of active infection, as well as current oncological disease, were excluded. At the time of inclusion and after 8 months, participants underwent a bicycle stress test (ergometry), respectively. In between, they were asked to perform at least 75 min/week of vigorous or 150 min/week of moderate intensity endurance training to assess the effect of exercise on physical performance after 8 months. Blood was drawn at baseline and at scheduled visits once every two months.

The study was carried out in adherence to the Declaration of Helsinki of the World Medical Association and its later amendments. The study protocol was approved by the Ethics Committee of the Medical University of Vienna (EC-number: 1830/2013) and written informed consent was obtained from all subjects. The study is registered at Clinical Trails (Influence of Physical Activity on Promising Atherosclerosis Biomarkers, NCT02097199).

### 2.2. Assessment of Patient Medical History and Definitions

Documentation of patient medical history and a detailed physical examination preceded study inclusion. Height was determined, and weight, body water, body muscle mass, and body fat were measured with a diagnostic scale (Beurer BG 16, Beurer GmbH, Ulm, Germany). Body surface was calculated according to the DuBois formula: body surface (m^2^) = 0.007184 × height (cm) 0.725 × weight (kg) 0.425). Overweight (body mass index > 25.0 kg/m^2^), hypertension (systolic blood pressure > 140 mmHg ± diastolic blood pressure > 85 mmHg at rest/antihypertensive therapy), dyslipidemia (statin therapy), diabetes mellitus (HbA1c > 6.5 rel%/diabetes medication), smoking status, known CVD (myocardial infarction, percutaneous coronary intervention, coronary artery bypass graft, stroke) and positive family history of first-degree relatives (mother or father) for CVD were assessed.

### 2.3. Bicycle Stress Test (Ergometry) and Continous Endurance Training

At the beginning of the study, participants completed a bicycle stress test (ergometry) to assess their initial performance level and to define their individual target heart frequency (HF) for endurance training. Therefore, the Karvonen formula with an intensity level of 65–75% for moderate and 76–93% for vigorous intensity was used: HF_training_ = HF_rest_ + (HF_max_ − HF_rest_) × intensity level. Thereby, the intensity level was chosen to support an aerobic to partially aerobic metabolism and improvement in endurance.

All participants were encouraged to engage in regular physical activity defined as 75 min/week of vigorous or 150 min/week of moderate intensity endurance training. While additional strength training was allowed (but not mandatory) within the calculated training pulse, no other specifications limited the individual choice of sports. This approach aimed at encouraging enjoyment of exercise, sustaining motivation, while not excluding subjects with lesser agility or articular trouble. A second ergometry completed the study period of 8 months to objectify the change in performance level. Each bicycle stress test was monitored by electrocardiography using the same system (Ergometer eBike comfort, GE Medical Systems, Freiburg, Germany). In agreement with current Austrian and European guidelines, the test was started with an intensity of 25 watts which was increased every 2 min by additional 25 watts. Blood pressure and heart rate were recorded every 2 min. Subjects were instructed to maintain 50–70 revolutions/min until exhaustion, which terminated the stress test. The target performance was calculated according to gender-specific formulas using body surface, sex and age for men: performance (W) = 6773 + 136.141 × body surface − 0.916 × body surface × age; and women: performance (W) = 3933 + 86.641 × body surface − 0.346 × body surface × age. An individual target performance of 100% represents the performance of an untrained population.

### 2.4. Blood Collection and Routine Laboratory Testing

At scheduled visits every two months, blood was drawn from the cubital vein using a tube-adapter system. Routine laboratory parameters were immediately analyzed after blood draw. Research samples were processed and stored by the central biobank until analysis. In summary, five serum samples per participant in two-month intervals were used for the following analyses.

### 2.5. Measurement of cfDNA and DNase Activity

CfDNA was measured using the fluorescent DNA-binding dye Sytox Green (ThermoFisher, Vienna, Austria). Serum was diluted 1:10 in PBS containing 0.1% BSA and 5 mmol/L EDTA and then mixed with an equal volume of 2 µM Sytox Green. Lambda DNA (ThermoFisher, Vienna, Austria) was diluted to a final concentration of 250 ng/mL of a 7-point two-fold standard curve. Data acquisition was performed using a Glomax microplate reader (Promega) at an excitation wavelength of 480 nm and an emission wavelength of 520 nm. After subtraction of the blank, fluorescence intensities of plasma samples were calculated against the standard curve.

To determine total DNase activity, serum samples were subjected to a single radial enzyme diffusion assay as previously described [29]. Salmon testes DNA (Sigma-Aldrich, St. Louis, MO, USA) was dissolved in assay buffer containing divalent cations and a DNA-binding fluorescent dye (35 mM Tris–HCl, pH 7.8, 20 mM MgCl_2_, 2 mM CaCl_2_, 2.5 × SYBR Safe (Invitrogen, Vienna, Austria)) at a concentration of 100 µg/mL as substrate for DNases. The solution was heated to 50 °C for 10 min and mixed with an equal volume of 2% ultra-pure agarose (Invitrogen, Carlsbad, CA, USA). After solidification in plastic trays, 2 µL sample or recombinant DNase1 standard (Dornase alfa, Roche) was loaded into 1 mm wide wells. Gels were incubated for 20 h at 37 °C. Remaining fluorescence was recorded with a Fusion FX imaging system (Vilber). DNase activity of samples was calculated in comparison to a 6-point standard curve starting at 25 mU/mL.

### 2.6. Statistical Analysis

Data distribution was analyzed via box-and-whiskers plots and the Kolmogorov–Smirnov test. Continuous, normally distributed data are presented as mean ± standard deviation (SD) and non-parametric data are described by median and interquartile range (IQR) in tables. Friedman test was conducted to investigate differences over the 8-month observation period. Unpaired or paired *t*-tests as well as Wilcoxon matched-pairs signed rank tests were applied to compare two groups, as appropriate. Spearman’s rho analysis was performed to calculate correlations involving non-parametric data. Wald’s backward multiple linear regression analysis was performed to predict cfDNA levels and DNase activity, respectively, adding all parameters, which have significantly correlated in Spearman’s rho analysis (*p* < 0.05) and setting the level of significance for inclusion into the model to *p* < 0.1. Multiple testing was corrected with the Bonferroni–Holm method, giving corrected and uncorrected *p*-values in the Appendix A.

Statistical analysis was performed using SPSS 26.0 (IBM Corp. Released 2019. IBM SPSS Statistics for Windows, Version 26.0. Armonk, NY, USA: IBM Corp) and GraphPad Prism version 9.0.0 for Windows (GraphPad Software, San Diego, CA, USA, www.graphpad.com).

## 3. Results

### 3.1. Patient Baseline Characteristics

We studied 98 subjects (39 women and 60 men) recruited out of the staff of the Austrian Federal Ministry of Defence. In the total study population, the prevailing cardiovascular risk factors were overweight (65.9%), a positive family history for CVD (44.9%), hypertension (32.7%) and smoking (20.4%), as previously reported [8]. On average, individuals suffered from 2.6 cardiovascular risk factors and 16 participants had a diagnosis of coronary heart disease. A breakdown of the number of risk factors per participant is given in Figure A1. Altogether, performance changed by 7.8 ± 9.1% after 8 months. Table 1 summarizes baseline demographical and laboratory parameters for the total study population and with respect to different athletic backgrounds of subjects and performance categories. As not all study participants could equally improve, a gain of 3% was considered clinically significant, serving as a threshold for classification into the following groups: non-sportive, gain < 2.9% (group 1, n = 9); non-sportive, gain > 2.9% (group 2, n = 32); sportive, gain < 2.9% (group 3, n = 18); sportive, gain > 2.9% (group 4, n = 39). In total, 71 individuals achieved a performance gain > 2.9% and 27 did not.

Initially sportive subjects (groups 3 + 4) had significantly higher body muscle tissue (35.6 ± 3.9 vs. 33.5 ± 3.9%, *p* = 0.014) and body water (54.1 ± 6.1 vs. 49.9 ± 4.5%, *p* < 0.001) as well as lower body fat (27.5 ± 10.9 vs. 32.1 ± 6.2%, *p* = 0.018) than initially non-sportive participants (group 1 + 2). Body mass index did not differ significantly between groups (28.3 ± 5.0 vs. 26.9 ± 3.4 kg/m2, *p* = 0.105). The mean performance change in group 1 was −2.7 ± 4.3%, in group 2 12.2 ± 7.1%, in group 3 −3.8 ± 4.9% and in group 4 12.1 ± 5.6%. While there was no difference in systolic blood pressure (BP, Figure 1A) or the heart rate (Figure 1C) at rest comparing baseline with 8 months, diastolic BP was significantly lower at the end of the trial (*p* = 0.007, Figure 1B). Analyzing the groups separately revealed a difference of diastolic blood pressure at rest for group 2 who were able to improve in performance (Figure 1E). The same observations could be made for maximum heart rate, systolic BP and diastolic BP during cardiopulmonary exercise (Figure A2) comparing baseline with follow-up after 8 months.

### 3.2. Predictors of cfDNA Levels and DNase Activity at Baseline

CfDNA levels and DNase activity were measured in two-month intervals covering all five clinical visits of the trial protocol. Due to the variable athletic conditions at the start of the study, baseline values of initially non-sportive and sportive groups were compared. No significant difference of cfDNA (*p* = 0.212, Figure 2A) or DNase activity (*p* = 0.318; Figure 2B) was detected. Comparing two groups dichotomized based on performance gain after training intervention did not reveal any changes after 8 months regarding cfDNA (*p* = 0.281, Figure 2C) and DNase activity (*p* = 0.311, Figure 2D).

In an exploratory approach, baseline cfDNA levels and DNase activity were correlated with other clinical parameters (Table A1). In the total population, cfDNA levels correlated positively with soluble urokinase plasminogen activator receptor (suPAR, *p* = 0.019), heart-type fatty-acid-binding protein (H-FABP, *p* = 0.048), body muscle mass (*p* = 0.029), hematocrit (*p* = 0.041), uric acid (*p* = 0.014), total amylase (*p* = 0.031), γ-glutamyltransferase (gGT, *p* = 0.006), ferritin (*p* = 0.047) androstendion (*p* = 0.006) and negatively with soluble receptor for advanced glycation endproducts (sRAGE, *p* = 0.039) and osteoprotegerin (*p* = 0.029). Backwards multiple linear regression analysis was used to stepwise eliminate parameters not improving the model to predict cfDNA levels, as defined by a *p*-value ≥ 0.1. The final model significantly predicted cfDNA levels (F(6, 89) = 9.263, *p* < 0.001, R2 = 0.384 (adjusted 0.343)) explaining 38.4% of the variance, and is presented in Table 2. The multiple correlation coefficient was 0.620 representing a good level of prediction.

The same approach was implemented for baseline DNase activity, which correlated positively with complement factor D (CFD, *p* = 0.008), aspartate aminotransferase (ASAT, *p* = 0.043), ferritin (*p* = 0.014), and negatively with osteoprotegerin (*p* = 0.047), platelet count (*p* = 0.016), LDL-cholesterol (*p* = 0.003) and apolipoprotein B (*p* = 0.021). Again, backwards multiple linear regression analysis was performed, which identified complement factor D (*p* = 0.053), ferritin (*p* < 0.001) and LDL-cholesterol (*p* = 0.005) as significant predictors (*p*-value < 0.1) of baseline DNase activity (F(3, 92) = 10.039, R2 = 0.247 (adjusted 0.222, Table 3)). The level of prediction was moderate, with an R = 0.497.

### 3.3. Influence of Physical Exercise on Levels of cfDNA and Activity of DNase

The trends of cfDNA levels and DNase activity over the full 8-month-period are presented in Figure 3. In the total study population, levels of cfDNA (*p* = 0.025, n = 82) and activity of DNase (*p* = 0.002, n = 80) differed significantly between time points as calculated by Friedman test. Comparison of baseline to 8 months revealed a prominent difference for both cfDNA (*p* = 0.002, Figure 3A,C) and DNase activity (*p* = 0.001, Figure 3B,C).

To identify whether the change in cfDNA and DNase activity after 8 months of exercise was driven by one of the performance categories defined above, the four groups were separately analyzed (Figure 4, Table A2 and Table A3). The groups not able to improve in performance also did not reveal any significant changes in cfDNA levels or DNase activity after correction for multiple testing (Figure 4A,C). Notably, the initial rise in DNase activity comparing baseline with month 2 was significantly different before correction (uncorrected *p*-value < 0.05, Figure 4C, Table A3). In contrast, group 2, whose members were untrained at the beginning but managed to achieve a performance gain, showed significantly higher DNase activity at study end compared to baseline (corrected *p*-value = 0.039) and also significantly lower cfDNA levels (corrected *p*-value = 0.024; Figure 4B). In accordance, even initially sportive participants who were still able to improve revealed a significant decrease in cfDNA at 8 months compared to baseline (corrected *p*-value = 0.005) and compared to 6 months (corrected *p*-value < 0.001, Figure 4D). DNase activity increased significantly between baseline and final follow-up (corrected *p*-value = 0.002, Figure 4D).

Although the capacity to degrade cfDNA was increased after 8 months of successful exercise, as presented by the ratio between cfDNA and DNase activity (Figure 5B,D, Table A5), no correlations between cfDNA levels and DNase activity could be observed at 8 months (Figure A3, Table A4) or the fold changes (data not shown).

## 4. Discussion

Physical activity represents a non-pharmacological intervention, benefitting cardiovascular health and complementing primary and secondary prevention. In the present study, we investigated the effects of consequent endurance training on DNase activity degrading cfDNA in the circulation. We observed endurance exercise as an enhancer of plasma DNase activity, exclusively in groups that reached a significant performance gain over the observational period of 8 months.

While there are many studies reporting immediate, short-term effects of high-performance endurance training on neutrophil activation, NET formation and even some on DNase activity [14,34,35,36,37], real-world cohort data on long-term effects with observation periods covering several months are scarce.

In the field of CVD, high cfDNA is regarded as a biomarker of disease severity and predictor of patient outcome [29,38,39]. However, levels may vary greatly between individuals in healthy and disease states; hence, it is not surprising that we could not find any significant differences comparing baseline cfDNA levels between sportive and non-sportive study participants. The cellular origins of circulating cfDNA are thereby mostly dependent on the concomitant circumstances or comorbidities [40]. With regards to sportive activity, the proposal that cfDNA is an early bystander marker of muscle damage [41] resulting from repetitive and sustained tissue micro-trauma was challenged by the rather rapid and transient peak patterns, which were not mirrored by other markers of muscle damage [34]. The extent of cfDNA release was thereby observed to be dependent on duration and intensity of exercise, neglecting the potential of long-lasting effects of regular exercise on cfDNA. Peripheral levels were found to immediately peak in response to acute exercise [14,16,42,43], subsiding between two hours [16] and 48 h [43]. Physical activity induces a multisystem stress response, which modulates immunity [7,44,45]. Secretion of endocrine hormones as well as of pro- and anti-inflammatory cytokines, next to an increasing production of reactive oxygen species, challenge the body’s effort to maintain physiological hemostasis [11]. These conditions promote accelerated release of neutrophils from the bone marrow, and their subsequent entrance into circulation is facilitated by catecholamines increasing permeability of the endothelium [13].

Notably, in this study, blood was not drawn immediately after intense exercise, but in regular intervals after the instruction to perform consequent, guideline-directed endurance training. Accordingly, we observed the homeostatic balance rather than provoked release of DNA caused by acute stress or inflammatory responses, showing an ultimate decline in cfDNA levels in the total cohort and especially in participants who achieved a performance gain. This could be indicative of reduced release of DNA and adaption to regular exercise. Sources might be a combination of neutrophils undergoing NET formation [34], extracellular vesicles from various tissues with DNA as cargo [46] and exercise-induced muscle damage [41]. However, potential medical preconditions will most likely influence basal cfDNA levels, enhancing the release from one or more of these sources. For example, a third of the study population suffered from arterial hypertension, which was recently shown to promote NET formation based on elevated levels of DNA–myeloperoxidase complexes and citrullinated histone H3 [47]. Although backwards multiple linear regression analysis could identify several factors predicting cfDNA levels at baseline, a considerable proportion of the total variance remains unexplained. Consequently, all parameters were normalized to baseline to monitor the individual time course during the 8-month study period and enable proper comparisons.

Importantly, not only the release of cfDNA but also its elimination from the circulation determines plasma concentrations. Higher DNase activity is regarded as a beneficial mediator reducing the pro-inflammatory and pro-thrombotic potential of cfDNA and NETs by timely degradation [48]. As major waste-management endonucleases, different DNases exist to catalyze hydrolysis of DNA in circulation or intracellularly, e.g., during apoptosis [49]. DNase is a promising tool in the treatment of cystic fibrosis [50], holds protective autoimmune [51] and anti-metastatic properties [52] upon increased activity, and predicts a favorable long-term outcome in patients after ST-segment elevation myocardial infarction [30,38]. Insufficient degradation of cfDNA due to inadequate activity of DNase1 provokes hemostatic turbulences and tissue damage, as well as thrombosis or immune complex formation [53]. At first, we speculated that increased DNase activity might result from the body’s effort to counter regulate exercise-provoked release of cfDNA and sub-clinical chronic inflammation, as proposed by previous research [34,37]. However, our data show a disproportionally higher increase in DNase activity, as evidenced by the cfDNA/DNase ratio, mirroring an adjusted balance in response to consequent exercise with actual performance gain. This observation could suggest that physical activity reduces the relative exposure to cfDNA by an increase in total DNase activity, limiting the immunogenic and pro-inflammatory risks. Importantly, neither absolute levels of cfDNA and DNase nor the relative change correlated, indicating once more an uncoupled constant increase in DNase activity, not exclusively in response to acute DNA release.

DNase1 can be produced by several tissues, foremost by liver and intestines, while DNase1L3 is predominantly released by cells of the myeloid lineage [54]. Recently, it has been shown that increased total cholesterol levels above 200 mg/dL attenuate a timely DNase response to pro-inflammatory stimuli by inducing endoplasmic reticulum (ER)-stress [55]. Although the exact mechanism of downregulated DNase activity in the circulation is yet unexplored, our data show a significant inverse correlation between LDL-cholesterol and DNase activity. Notably, LDL-cholesterol predicted DNase activity in multiple linear regression analysis (Table 3) and showed a declining trend comparing baseline and 8-month follow-up in the total cohort (data not shown, *p* = 0.071). Altered cholesterol levels could be a result of an overall healthier lifestyle adopted during the study period. Whether DNase activity would increase in response to more intense secondary prevention treatments (e.g., PCSK-9 inhibitors [56], anti-inflammatory treatment) should be investigated in further studies.

Altogether, our study emphasizes the benefits of regular exercise, as measured by performance during a bicycle stress test. A predominant sedentary lifestyle promotes adverse health outcomes, such as increasing all-cause mortality, CVD and cancer. Regular integration of physical activity might delay or prevent these detrimental conditions, along with improvement in mental and cognitive health, as well as incidental hypertension and type-2 diabetes mellitus [6]. In this regard, specifically the increase in DNase activity could be of major advantage in patients at cardiovascular risk. Notably, exercise seemed to stimulate a fast adaptive response, elevating DNase activity independent of current performance gain. This was particularly evident for group 3 (sportive, no gain), who showed an early spike comparing baseline and 2 months (uncorrected *p*-value < 0.05), potentially driven by initial over-keen fitness goals. Trained athletes hold a greater ‘fatigue resistance’, which enables them to tolerate increased strain, but also elevates the threshold required in order to improve [57], whereas untrained individuals are more likely to experience fast progress. In accordance, we speculated that a fold increase in DNase activity might reflect the individual adherence to a training schedule, which could potentially be used to monitor a patient’s compliance and provide motivational as well as specific training support.

The study lacks a clean control group, as recommendation and prescription of exercise is the standard of care in primary and secondary prevention of individuals at cardiovascular risk, which cannot be denied without infringement of ethical standards or compromise of the participants’ health. Subjects who could not improve their performance have likely not adhered to the exercise recommendations representing the controls (groups 1 and 3) in this study. Importantly, categorization was also supported by a decrease in diastolic blood pressure at rest and during cardiopulmonary exercise in groups that improved by exercising. Regardless of their initial fitness condition, more than two-thirds (72%) of the total population gained at least 3% in performance because of effective implementation of training recommendations. However, subdivision into four groups also reduced the sample size and any interpretation has to be made with caution. Documentation of, as well as time lag between, sportive activities and the study blood draws could reduce the remaining uncertainties and support the associations made in this discussion. The biological purpose and pathophysiological consequences of exercise-induced elevation in DNase activity remain unclear as the results of this study are overly observational and outcome is restricted to physical fitness. However, it was previously shown that a point mutation in DNase1 slightly decreasing enzymatic function was significantly associated with long-term cardiovascular and all-cause mortality [29]. We can only speculate about the clinical relevance in our cohort, as the study is not only too small, but the observation period is also too short to present such data. Further research is required to determine whether the exercise-provoked increase in DNase activity holds potential as interventional prescription.

In summary, physical activity might not only improve cardiorespiratory fitness of patients, but also increase the capacity to lower NET burden and pro-inflammatory signaling through an exercise-induced rise in DNase activity, benefitting patients in primary and secondary prevention of CVD.

## Figures and Tables

**Figure 1 biomedicines-10-02849-f001:**
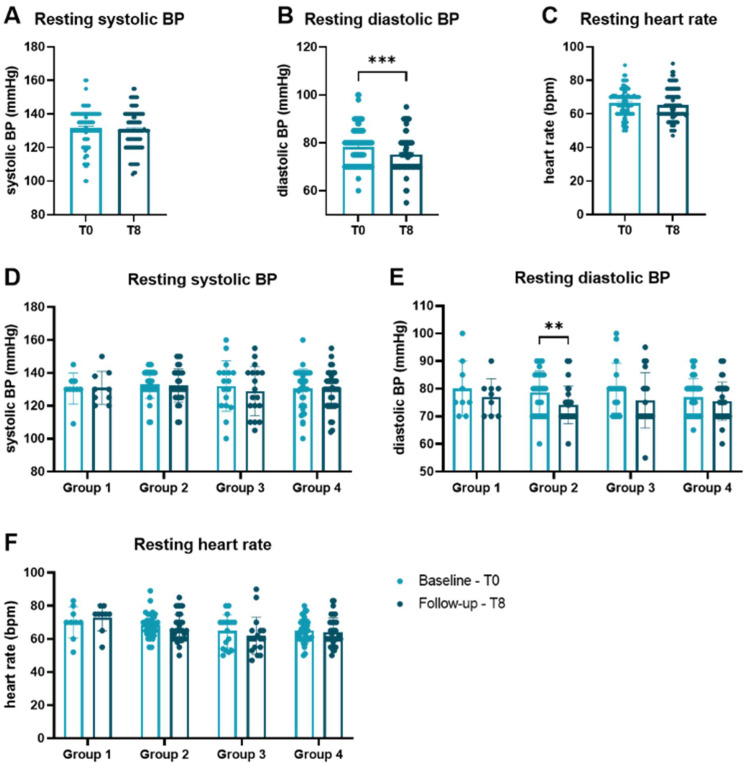
Vital parameters at rest at baseline and follow-up. Comparison of (**A**) systolic BP, (**B**) diastolic BP, and (**C**) heart rate measured at rest between baseline and 8 months. (**D**) Systolic BP, (**E**) diastolic BP, and (**F**) heart rate are presented according to performance categories. Groups were compared by Wilcoxon matched-pairs signed rank tests. Data are presented as scatter dot plots with mean ± SEM. ** *p* < 0.01, *** *p* < 0.001. BP blood pressure, bpm beats per minute, SEM standard error of the mean.

**Figure 2 biomedicines-10-02849-f002:**
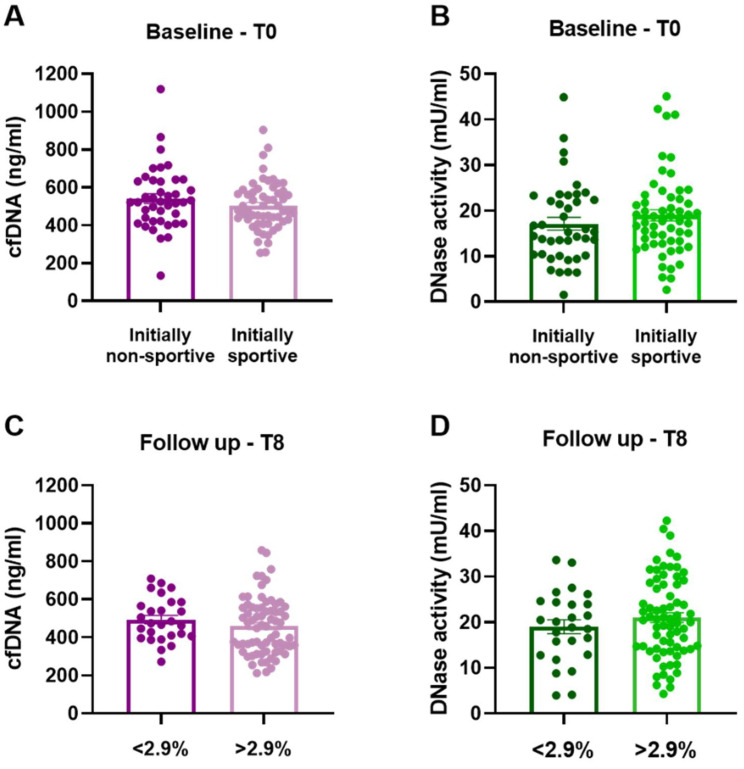
Levels of cfDNA and DNase activity at baseline and after 8 months of training intervention. (**A**) Baseline cfDNA levels, *p* = 0.212 and (**B**) baseline DNase activity, *p* = 0.318 of the initially non-sportive (n = 41) and sportive group (n = 57) group. 8-month (**C**) cfDNA levels, *p* = 0.281 and (**D**) DNase activity, *p* = 0.311, of groups with <2.9% (n = 27) and >2.9% (n = 71) performance gain. Data are presented as scatter dot plots with mean ± SEM, and were compared by unpaired two-tailed *t*-tests.

**Figure 3 biomedicines-10-02849-f003:**
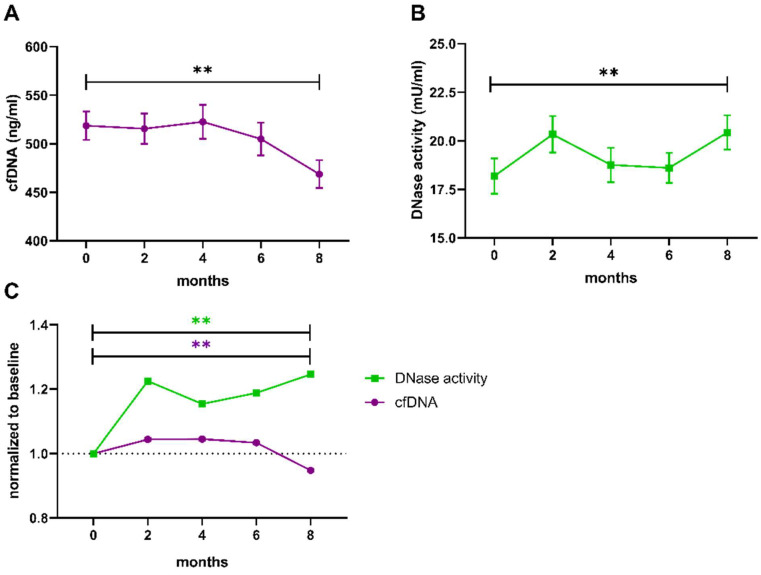
Trends of cfDNA levels and DNase activity during the 8-month trial period. Both parameters were measured in samples from baseline, after 2, 4, 6 and 8 months. (**A**) Levels of cfDNA and (**B**) DNase activity were compared between baseline and 8 months using a paired two-sample *t*-test. (**C**) Fold changes of cfDNA levels (magenta) and DNase activity (green) were calculated by normalization to the individual baseline. Data are presented as mean ± SEM. ** *p* < 0.01, cfDNA cell-free DNA, SEM standard error of the mean.

**Figure 4 biomedicines-10-02849-f004:**
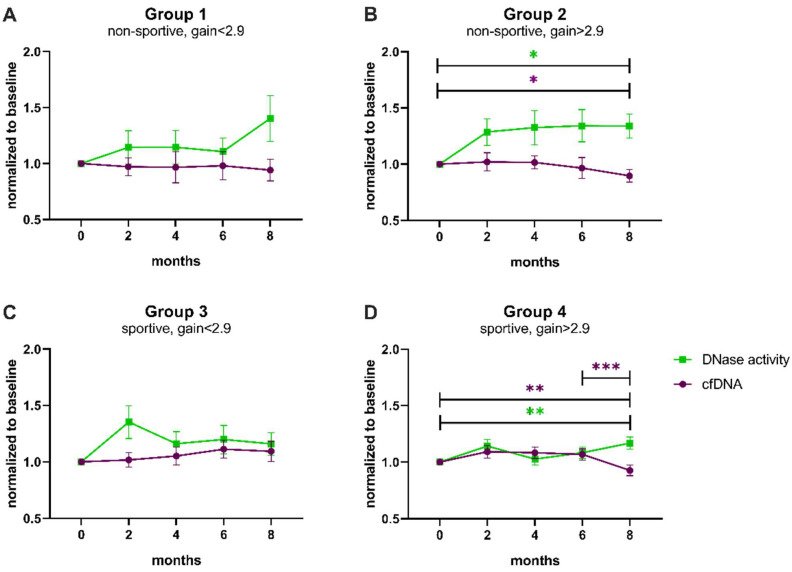
Trends of cfDNA levels and DNase activity normalized to baseline according to performance categories. (**A**) Non-sportive, gain < 2.9, (**B**) non-sportive, gain > 2.9, (**C**) sportive, gain < 2.9, (**D**) sportive, gain > 2.9. cfDNA levels (magenta) and DNase activity (green) were compared between baseline and 2 months, baseline and 8 months, and 6 and 8 months by paired *t*-tests, which were corrected using the Bonferroni–Holm method (Table A2 and Table A3). Data are presented as mean ± SEM. * *p* < 0.05, ** *p* < 0.01, *** *p* < 0.001, SEM standard error of the mean.

**Figure 5 biomedicines-10-02849-f005:**
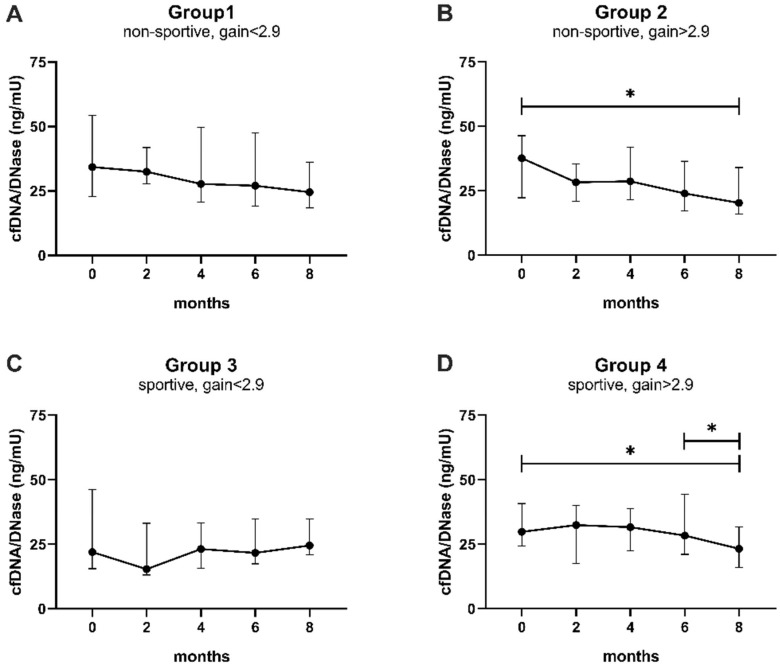
The capacity to balance cfDNA is increased in response to performance gain. (**A**) Non-sportive, gain < 2.9, (**B**) non-sportive, gain > 2.9, (**C**) sportive, gain < 2.9, (**D**) sportive, gain > 2.9. CfDNA levels were divided by DNase activity and compared between baseline and 2 months, baseline and 8 months, and 6 and 8 months by Wilcoxon matched-pairs signed rank tests, which were corrected using the Bonferroni–Holm method (Table A5). Data are presented as median ± IQR. * *p* < 0.05, IQR interquartile range.

**Table 1 biomedicines-10-02849-t001:** Cardiovascular risk factor profile, anthropometric, and laboratory data at baseline of all four groups. Data are presented as mean ± SD.

Parameters	Group 1 Non-Sportive Gain ≤ 2.9% (n = 9)	Group 2 Non-Sportive Gain > 2.9% (n = 32)	Group 3 Sportive Gain ≤ 2.9% (n = 18)	Group 4 Sportive Gain > 2.9% (n = 39)	Total Population (n = 98)
Age (years)	50.3 ± 6.1	48.6 ± 7.9	50.4 ± 6.5	49.1 ± 6.0	49.3 ± 6.7
BMI (kg/m^2^)	27.8 ± 4.2	28.5 ± 5.2	27.2 ± 3.8	26.8 ± 3.3	27.5 ± 4.2
Body fat (%)	33.9 ± 3.3	31.6 ± 6.7	26.8 ± 9.1	27.8 ± 11.8	29.4 ± 9.5
Body muscle (%)	32.4 ± 3.3	33.9 ± 4.1	34.3 ± 3.8	36.1 ± 4.0	34.7 ± 4.1
Body water (%)	48.6 ± 2.4	50.3 ± 4.9	53.8 ± 6.7	54.2 ± 5.9	52.3 ± 5.9
Performance baseline (%)	87.4 ± 9.9	88.8 ± 7.1	122.0 ± 16.8	116.0 ± 15.9	105.6 ± 19.7
Performance study end (%)	87.0 ± 9.1	101.0 ± 10.0	118.2 ± 18.0	128.2 ± 15.6	113.7 ± 20.0
Performance gain (%)	−2.7 ± 4.3	12.2 ± 7.1	−3.8 ± 4.9	12.1 ± 5.6	7.8 ± 9.1
Pack years	22.4 ± 21.4	18.9 ± 15.8	12.2 ± 9.2	16.3 ± 14.6	17.1 ± 14.9
Alcohol intake (units/week)	0.7 ± 1.0	2.8 ± 3.2	3.4 ± 4.0	3.2 ± 4.4	2.9 ± 3.8
Male sex (%)	44.4	53.1	61.1	71.8	61.2
Active smoking (%)	55.6	25.0	16.7	10.3	20.4
Cardiac history (%)	11.1	15.6	5.6	23.1	16.3
Diabetes mellitus (%)	11.1	3.1	5.6	0	3.1
Hypertension (%)	33.3	43.8	33.3	23.1	32.7
Dyslipidemia (%)	33.3	25.0	38.9	28.2	29.6
Overweight (%)	66.8	68.8	66.7	63.2	65.9
Positive family history (%)	66.8	43.8	50.0	38.5	44.9
Erythrocytes (T/L)	4.6 ± 0.4	4.8 ± 0.5	4.6 ± 0.4	4.7 ± 0.4	4.7 ± 0.4
Haemoglobin (g/dL)	13.3 ± 1.5	14.2 ± 1.5	13.8 ± 1.0	14.2 ± 1.2	14.0 ± 1.3
Sodium (mmol/L)	141 ± 2	141 ± 2	141 ± 2	142 ± 2	141 ± 1.7
Potassium (mmol/L)	4.2 ± 0.2	4.1 ± 0.2	4.2 ± 0.3	4.2 ± 0.2	4.2 ± 0.3
Creatinine (mg/dL)	0.8 ± 0.1	0.8 ± 0.2	0.9 ± 0.2	0.9 ± 0.2	0.9 ± 0.2
ASAT (U/L)	23 ± 4	26 ± 10	27 ± 7	24 ± 5	25 ± 7
Triglycerides (mg/dL)	154 ± 86	149 ± 100	111 ± 72	119 ± 62	131 ± 81
Cholesterol (mg/dL)	209 ± 54	200 ± 37	196 ± 29	201 ± 39	200 ± 38
HDL-cholesterol (mg/dL)	52 ± 19	56 ± 22	62 ± 12	60 ± 15	59 ± 17
LDL-cholesterol (mg/dL)	126 ± 50	117 ± 32	112 ± 29	116 ± 35	117 ± 34
HbA1c (rel.%)	5.5 ± 0.4	5.4 ± 0.8	5.5 ± 0.9	5.2 ± 0.3	5.3 ± 0.6
proBNP (pg/mL)	39 ± 27	59 ± 54	50 ± 35	32 ± 21	45 ± 39

ASAT Aspartat-Aminotransferase, HDL high-density lipoprotein, LDL low-density lipoprotein, HbA1c Hemoglobin A1C, proBNP N-terminal prohormone of brain natriuretic peptide.

**Table 2 biomedicines-10-02849-t002:** Backwards multiple linear regression analysis of cfDNA levels at baseline.

cfDNA	Standardized B	T	*p* Value	95% CI
suPAR, pg/mL	0.239	2.770	0.007	0.021; 0.129
H-FABP, ng/mL	0.234	2.669	0.009	9.073; 61.95
Body muscle, %	0.159	1.695	0.093	−0.984; 12.43
Osteoprotegerin, pmol/L	−0.199	−2.351	0.021	−23.42; −1.966
Uric acid, mg/dL	0.204	2.107	0.038	1.231; 41.89
Androstendion, ng/mL	0.200	2.335	0.022	7.246; 89.92

suPAR soluble urokinase plasminogen activator receptor, H-FABP heart-type fatty-acid-binding protein.

**Table 3 biomedicines-10-02849-t003:** Backwards multiple linear regression analysis of DNase activity at baseline.

DNase Activity	Standardized B	T	*p* Value	95% CI
CFD, pg/mL	0.181	1.957	0.053	0.000; 0.000
Ferritin, µg/L	0.338	3.670	<0.001	0.014; 0.048
LDL cholesterol, mg/dL	−0.261	−2.848	0.005	−0.114; 0.020

CFD complement factor D, LDL low-density lipoprotein.

## Data Availability

Data can be accessed upon reasonable request.

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
