# Peer review of "Physical Exercise Promotes DNase Activity Enhancing the Capacity to Degrade Neutrophil Extracellular Traps"

_biomedicines, 2022, doi:10.3390/biomedicines10112849_

Round 1

Reviewer 1 Report

Ondracek et al reported that physical activity might improve cardiovascular health by increasing DNase activity. This study was interesting. However, several results or information are insufficient. Unfortunately, I can’t suggest this paper is acceptable.

1.    The background is unclear. The introduction still largely undefined the importance of DNase activity in CVD.

2.    Healthy subjects were involved in this study, which reduced the clinical implications of the results. CVD cases are suggested.

3.    The exercise protocol is unclear. How to adjust the exercise prescription? Warm up? Cool down? training location? Who takes responsibility for prescript and conducting exercise training?

4.    No provide CPET results.

5.    This study include healthy subjects, however, the exercise test was performed by bicycle. A treadmill exercise testing is suggested.

6.    It is unclear the termination criteria for the exercise test. 

Author Response

Ondracek et al reported that physical activity might improve cardiovascular health by increasing DNase activity. This study was interesting. However, several results or information are insufficient. Unfortunately, I can’t suggest this paper is acceptable.

  1. The background is unclear. The introduction still largely undefined the importance of DNase activity in CVD.

We thank the Reviewer for the insight and have revised the introduction and tried to highlight the importance of DNase activity in CVD (page 2).

  1. Healthy subjects were involved in this study, which reduced the clinical implications of the results. CVD cases are suggested.

As stated in the manuscript, for this study, volunteers suffering from at least one cardiovascular risk factor (CV-RF) have been included: 1) Diagnosed chronic heart disease with prior myocardial infarction, CABG, PCI, or stroke, 2) a positive family history of first-degree relatives (mother or father) regarding cardiovascular disease or stroke, 3) the presence of one or more metabolic risk factors including overweight (BMI > 25), diabetes mellitus (HbA1c > 6.5% or the presence of antidiabetic medication), dyslipidemia (marked by statin intake), and arterial hypertension (resting SBP > 140 mmHg/resting DBP > 85 mmHg or the presence of antihypertensive medication), or 4) a positive smoking status.

To give a better overview of the CV-RF profile, we have included a histogram presenting the number of CV-RFs per subject (Figure A1). The figure is referenced in the section 3.1 on page 5. On average, individuals suffered from 2.6 CV-RFs. In detail, there was 1 individual without any CV-RFs, 16 subjects with 1, 31 subjects with 2, 26 subjects with 3, 12 subjects with 4, 7 subjects with 5, and 2 subjects with all of the above defined CV-RFs. Moreover, 16 participants had a diagnosis of coronary heart disease. These participants represent a real-world cohort at defined cardiovascular risk and though clinical implications might not be acute, they are potentially looming without preventive measures.

  1. The exercise protocol is unclear. How to adjust the exercise prescription? Warm up? Cool down? training location? Who takes responsibility for prescript and conducting exercise training?

At the beginning of the study, each individuals’ target heart frequency for subsequent endurance training was defined using the Karvonen formula as described in the methods section. We have expanded the methodological paragraph on the topic in section 2.3 to clarify the basis for the exercise prescription. The aim of the prescription was an improvement of performance as measured by a bicycle stress test. It was not of interest for the study objectives, which kind of sports or exercise was performed as long as within the calculated training pulse. Not restricting the individual choice of sports aimed at encouraging enjoyment of exercise while not excluding subjects with lesser agility.

  1. No provide CPET results.

We thank the Reviewer for the suggestion and included blood pressure and heart rate at rest as well as peak values during cardiopulmonary exercise testing summarized in the new Figures 1 and A2.

  1. This study include healthy subjects, however, the exercise test was performed by bicycle. A treadmill exercise testing is suggested.

The bicycle stress test is the standard method at our institution and harbors significant advantages to evaluate performance in a cohort at cardiovascular risk. The mean age of our participants was 49.3±6.7 years and as described above, they suffered on average from 2.6 CV-RFs. These characteristics are often associated with declining agility and besides, a bicycle stress test is possible despite potential articular trouble.

  1. It is unclear the termination criteria for the exercise test

In section 2.3. “Bicycle stress test (ergometry) and continous endurance training” we stated that “subjects were instructed to maintain 50-70 revolutions/min until exhaustion, which terminated the stress test.”

Reviewer 2 Report

Dear Authors

I review in regards to manuscript # biomedicines-1971630-peer-review-v1, entitled "Physical exercise promotes DNase activity enhancing the capacity to degrade neutrophil extracellular traps" which submitted to the biomedicines. The title of this study seems to be consistent with the journal. I think this paper will be better if some minor points are corrected.

Minor points

Q1: The 'comma' in the numbers given in lines 111 to 116 must be changed to a 'decimal point'.

Q2: The 'comma' in the numbers given in lines 135 to 136 must be changed to a 'decimal point'.

Q3: Statistical symbol 'p' given on line 177 and below should be corrected to 'italics'.

This paper has a well-planned design. In particular, this paper seems to have been well described without any particular errors.

Sincerely,

Author Response

Dear Authors

I review in regards to manuscript # biomedicines-1971630-peer-review-v1, entitled "Physical exercise promotes DNase activity enhancing the capacity to degrade neutrophil extracellular traps" which submitted to the biomedicines. The title of this study seems to be consistent with the journal. I think this paper will be better if some minor points are corrected.

Minor points

  1. Q1: The 'comma' in the numbers given in lines 111 to 116 must be changed to a 'decimal point'.
  2. Q2: The 'comma' in the numbers given in lines 135 to 136 must be changed to a 'decimal point'.
  3. Q3: Statistical symbol 'p' given on line 177 and below should be corrected to 'italics'.

This paper has a well-planned design. In particular, this paper seems to have been well described without any particular errors.

We thank the Reviewer for this very affirmative review and her/his eye for detail. We have changed the “commas” to “decimal points” and corrected the statistical symbol “p” to the proper italic “p”

Reviewer 3 Report

Title: Physical exercise promotes DNase activity enhancing the capacity to degrade neutrophil extracellular traps

This article seems well built and brings evidence of a phenomenon not yet fully understood and that certainly deserves further study.

Some hard points of revision are provided below

The authors would like to demonstrate the effects of 8-month endurance training on the capacity to degrade neutrophil extracellular traps

There are two big problems:

The first problem is that, unclear the endurance training was performed (thus: walking. Running etc; kilometers, speed) to better replicate this study

The second problem is that the control groups is missing (without any activity)

Therefore, this investigation cannot be publisher on a scientific journal

Author Response

This article seems well built and brings evidence of a phenomenon not yet fully understood and that certainly deserves further study. Some hard points of revision are provided below

The authors would like to demonstrate the effects of 8-month endurance training on the capacity to degrade neutrophil extracellular traps. There are two big problems; Therefore, this investigation cannot be publisher on a scientific journal:

  1. The first problem is that, unclear the endurance training was performed (thus: walking. Running etc; kilometers, speed) to better replicate this study

At the beginning of the study, each individuals’ target heart frequency for subsequent endurance training was defined using the Karvonen formula. As suggested by Reviewer 1, we have expanded the methodological paragraph on the topic in section 2.3 to clarify the basis for the exercise prescription. However, the aim of the prescription (all participants were encouraged to engage in regular physical activity defined as 75 min/week of vigorous or 150 min/week of moderate intensity endurance training) was an improvement of performance as measured by a bicycle stress test. It was not of interest for the study objectives, which kind of sports or exercise was performed as long as within the calculated training pulse. Not restricting the individual choice of sports aimed at encouraging enjoyment of exercise while not excluding subjects with lesser agility or articular trouble. We have now highlighted this approach in the methods section.

  1. The second problem is that the control groups is missing (without any activity).

We agree with the Reviewer that this interventional study is missing a true control group. However, the recommendation and prescription of exercise is standard of care in primary and secondary prevention of individuals at cardiovascular risk and can thus not be denied without infringement of ethical standards, or compromise of the participants’ health. By measuring performance at the beginning and end of the study, we were able to classify the study subjects into four groups based on der initial fitness and the level of improvement. Participants who could not improve their performance have likely not adhered to the exercise recommendations representing the controls (groups 1 and 3) in this study. To acknowledge this imposed shortcoming we have added the issue to the discussion.

Reviewer 4 Report

This is an interesting report on a well exploited field. The paper has a decent language level, even if some mistakes should be corrected here and there (work for copy editors).

Nevertheless I have some concerns:

Figure 1 shows that data differences, if any, are very small and well inside the range of data distribution (at least in the box and whiskers plots shown), have the claimed differences from the normalized data sets (do those data sets follow a normal distribution? J.M.Bland, D.G.Altman, BMJ. 1996; 312: 770) a clinical (i.e. real world) significance?

Data are presented as mean ± SD, mean ± SEM, or median ± IQR; I believe the authors should select only mean ± SD for a clear flow.

BP measurements are never shown. Since the physical exercise should reduce BP, it would be interesting to display those values, too.

Tables A3 and A5 should be made clearer: the order of presentation in the rows is quite clumsy.

Reference section is not very updated, since the the publication year mean ± SD is 2014.73 ± 6.32, even if the max frequency is for years 2021, 2020 and 2018.

Last but not least there is a paper the authors should cite, about long-term exercise effect on some parameters here discussed:

Valeria Oliveira de Sousa B, de Freitas DF, Monteiro-Junior RS, Mendes IHR, Sousa JN, Guimarães VHD, Santos SHS. Physical exercise, obesity, inflammation and neutrophil extracellular traps (NETs): a review with bioinformatics analysis. Mol Biol Rep. 2021 May;48(5):4625-4635. doi: 10.1007/s11033-021-06400-2. Epub 2021 May 20. PMID: 34014471.

Author Response

This is an interesting report on a well exploited field. The paper has a decent language level, even if some mistakes should be corrected here and there (work for copy editors).

Nevertheless I have some concerns:

  1. Figure 1 shows that data differences, if any, are very small and well inside the range of data distribution (at least in the box and whiskers plots shown), have the claimed differences from the normalized data sets (do those data sets follow a normal distribution? J.M.Bland, D.G.Altman, BMJ. 1996; 312: 770) a clinical (i.e. real world) significance?

To our knowledge, this is the first study in individuals at defined cardiovascular risk and not in an acute setting to investigate the balance between DNase activity and cell-free DNA. To date, we can only speculate about the clinical relevance based on data from acute myocardial infarction [1] or cardiac arrest patients [2]. However, it was shown that a point mutation in DNase1 slightly decreasing enzymatic function was significantly associated with long-term cardiovascular and all-cause mortality [3]. We are convinced that our study can be a basis for future trials focusing on patient outcome. Unfortunately, our study is not only too small but the observation period is too short to present such data. To acknowledge the Reviewers concerns, we have extended the discussion on limitations in our manuscript. 

  1. Data are presented as mean ± SD, mean ± SEM, or median ± IQR; I believe the authors should select only mean ± SD for a clear flow.

We thank the Reviewer for pointing out inconsistencies in our data presentation. We have changed the style of Figure 2 (which was previously Figure 1) to scatter dot plots showing individual data points rather than median ± IQR, as data are normally distributed and were compared by tests for parametric data. Concerning Figure 4 and legend 4, data are actually non-parametric and were therefore compared with Wilcoxon matched-pairs signed rank test and not as indicated by paired t-tests. This error was corrected, but accordingly, data are presented as median ± IQR. We hope the Reviewer agrees with our approach.

  1. BP measurements are never shown. Since the physical exercise should reduce BP, it would be interesting to display those values, too.

As suggested by the Reviewer, we have included blood pressure measurements as well as heart rate at rest and peak values during cardiopulmonary exercise as additional figures (Figure 1, Suppl. Figure A2). Peak diastolic blood pressure during the bicycle stress test as well as diastolic blood pressure at rest were significantly lower after 8 months of exercise and especially driven by groups with a performance gain.

  1. Tables A3 and A5 should be made clearer: the order of presentation in the rows is quite clumsy.

The order of presentation originates from the Bonferroni-Holm correction for multiple testing, where p-values are sorted from smallest to largest to be multiplied by the correction factor. The lowest p-value is multiplied by the total number of comparisons which is 3 in our case, the second lowest by 2, and the highest by 1. However, we have rearranged the data in the first columns to always depict the same order of time points to make the table clearer.

  1. Reference section is not very updated, since the the publication year mean ± SD is 2014.73 ± 6.32, even if the max frequency is for years 2021, 2020 and 2018.

We thank the Reviewer for making the effort to analyze our reference list. However, in agreement with citation guidelines, we have tried to cite original research articles whenever possible as compared to contemporary reviews. Nevertheless, we have also taken care to include the newest possible references for the revision of the introduction, which was suggested by Reviewer 1 (page 2).

  1. Last but not least there is a paper the authors should cite, about long-term exercise effect on some parameters here discussed:

Valeria Oliveira de Sousa B, de Freitas DF, Monteiro-Junior RS, Mendes IHR, Sousa JN, Guimarães VHD, Santos SHS. Physical exercise, obesity, inflammation and neutrophil extracellular traps (NETs): a review with bioinformatics analysis. Mol Biol Rep. 2021 May;48(5):4625-4635. doi: 10.1007/s11033-021-06400-2. Epub 2021 May 20. PMID: 34014471.

We agree with the Reviewer that this paper makes an excellent addition to our reference list and have inserted it into the last paragraph of the introduction.

Round 2

Reviewer 1 Report

My questions had been well addressed with this revision. This submission is acceptable. 

Reviewer 3 Report

Unfortunately the training program was not well designed therefore cannot be replicated anyway the control group is missing 

Reviewer 4 Report

I appreciated the efforts made to improve the paper. My opinion is that the paper may be now considered for publication in the present form